# Segmentation-Consistent Probabilistic Lesion Counting

**Julien Schroeter**[*1]                                                   JULIEN@CIM.MCGILL.CA

**Chelsea Myers-Colet**[*1]                                           CMYERS@CIM.MCGILL.CA

**Douglas L. Arnold**[2]                                         DOUGLAS.ARNOLD@MCGILL.CA

**Tal Arbel**[1]                                                         ARBEL@CIM.MCGILL.CA

[1] *Centre for Intelligent Machines, McGill University, Montreal, Canada*

[2] *Montreal Neurological Institute, McGill University, Montreal, Canada*

Editors: Under Review for MIDL 2022

## Abstract

Lesion counts are important indicators of disease severity, patient prognosis, and treatment efficacy, yet counting as a task in medical imaging is often overlooked in favor of segmentation. This work introduces a novel continuously differentiable function that maps lesion segmentation predictions to lesion count probability distributions in a consistent manner. The proposed end-to-end approach—which consists of voxel clustering, lesion-level voxel probability aggregation, and Poisson-binomial counting—is non-parametric and thus offers a robust and consistent way to augment lesion segmentation models with post hoc counting capabilities. Experiments on Gadolinium-enhancing lesion counting demonstrate that our method outputs accurate and well-calibrated count distributions that capture meaningful uncertainty information. They also reveal that our model is suitable for multi-task learning of lesion segmentation, is efficient in low data regimes, and is robust to adversarial attacks.

**Keywords:** Lesion Counting, Lesion Segmentation, Multi-task Learning, Robustness.

## 1. Introduction

Lesion counts are of great clinical importance. First, the number of lesions can be used as a metric for gauging the severity of pathologies such as Multiple Sclerosis (MS) (Khoury et al., 1994; Dworkin et al., 2018) and Alzheimer's disease (Bancher et al., 1993). Second, lesion counts can be an important biomarker for disease prognostic. For example, benign mole counts were found to be a key indicator of the risk of melanoma (Grob et al., 1990; Gandini et al., 2005). Similarly, the number of cerebral hemorrhages present in stroke patients (Greenberg et al., 2004) and the number of lesions in MS patients (Brex et al., 2002; Chung et al., 2020) have been shown to be correlated with the degree of future impairment. Finally, the number of lesions is also a common metric for evaluating treatment efficacy in, e.g., acne (Lucky et al., 1996, 1997) and MS (Rudick et al., 2006; Kappos et al., 2007).

Previous works in deep learning have mainly focused on segmenting lesions and, while great strides have been made in this area (Razzak et al., 2018; Lundervold and Lundervold, 2019), very few have tackled lesion counting, despite its clinical relevance. In an attempt to fill this gap, we derive a novel parameter-free *function* that can be leveraged to augment lesion segmentation models (Ronneberger et al., 2015; Isensee et al., 2018) with probabilistic lesion counting capabilities. Our method offers a unique and, most importantly, differentiable way of binding a segmentation map (function input) to a count distribution (function output) thus ensuring consistency, interpretability and robustness—key characteristics for deploying deep learning models in medical applications.

---

[*] Contributed equally

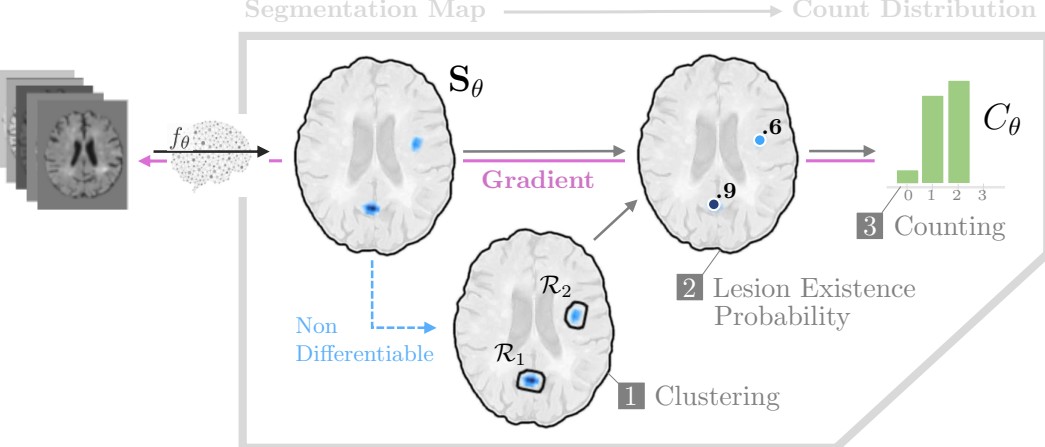

Figure 1: Method overview. Boxed numbers refer to subsections in Section 3.1 (3.1.x).

## 2. Related Work

**Lesion Counting** is often performed as a post-processing operation applied to lesion segmentation outputs; typically by 1) binarizing the segmentation map through thresholding before 2) counting the number of connected components on it. Variations of this framework are used to compute counting-based metrics in lesion segmentation challenges (Commowick et al., 2021; Timmins et al., 2021) or infer MS lesion counts (Dworkin et al., 2018). Though simple to deploy, the non-differentiability of this counting approach makes it unsuitable for gradient-based training. Naturally, several parametric models have also been proposed recently: e.g., deep count regression model (Dubost et al., 2020; Han et al., 2021), patch-wise classification model (Betz-Stablein et al., 2021). However, while differentiable, these models are inherently less interpretable and less robust due to their highly-parametrized nature.

In contrast, our proposed segmentation-based counting function is both *differentiable* and *non-parametric* thus offering a robust, interpretable and backpropagatable alternative.

**Multi-task Learning** is a common training scheme involving the simultaneous learning of multiple related tasks (Ruder, 2017). In medical imaging, many approaches have been proposed to combine pixel-level (segmentation) and image-level (classification) labels. Overall, the main modeling difference resides in the number of parameters shared among tasks. For instance, recent works have found success in placing a classification head at the bottleneck layer of a UNet (Wang et al., 2018; Le et al., 2019; Amyar et al., 2020; Zhou et al., 2021). Of course, a greater overlap in shared parameters can also be implemented, thus implicitly imposing a higher constraint on the segmentation network to extract relevant representations for the classifier. For example, Mlynarski et al. (2019) improved performance on brain tumor segmentation when classifying diseased brains at the second-to-last layer of a UNet, while Chen et al. (2019) leveraged both local bottleneck and top-level representations for melanoma detection. Finally, downstream classification models add layers directly on top of the segmentation output (Yu et al., 2016; Al-Masni et al., 2020; Khan et al., 2020, 2021).

In contrast, the counting and segmentation streams of our approach share *all* model parameters. We posit that this one-to-one correspondence ensures greater consistency between the predictions of the two tasks (see Section 4.2).

## 3. Method

In this work, we propose a novel probabilistic and continuously differentiable *function* that coherently maps lesion segmentation outputs to lesion count distributions. Most notably, our mapping is parameter-free thus ensuring a hard *consistency* between segmentation outputs and count predictions—unlike standard parametric multi-task approaches (Wang et al., 2018; Mlynarski et al., 2019; Tschandl et al., 2019; Al-Masni et al., 2020) where the segmentation and classification heads might produce contradictory results.

### 3.1. Poisson-binomial Lesion Counting

The starting point of this work is set in a framework that is ubiquitous in medical instance segmentation (Pehrson et al., 2019; Taghanaki et al., 2021): a segmentation model $f_\theta$ with trainable parameters $\theta$ maps an input image $\mathbf{X}$ to a segmentation output $\mathbf{S}_\theta$ of the same size—i.e., $\mathbf{S}_\theta = f_\theta(\mathbf{X})$. More importantly, it is assumed that the value of each voxel $v_i$ of $\mathbf{S}_\theta$ is an estimate of the probability of the corresponding voxel of $\mathbf{X}$ belonging to a lesion. In other words, the model $f_\theta$ estimates $p(V_i = 1 | \mathbf{X})$, where $V_i$ is the random variable indicating whether the voxel $v_i$ is part of a lesion. (For clarity, we now drop the dependence on $\mathbf{X}$.)

In this section, we present how a lesion segmentation map $\mathbf{S}_\theta$ can be mapped to a discrete count distribution $C_\theta$ that fully captures the number of instances in the input image—according to the segmentation model $f_\theta$. The approach is summarized in Figure 1.

#### 3.1.1. Lesion Candidate Identification

Counting the number of lesions in a segmentation map $\mathbf{S}_\theta$ requires the identification of individual lesions from the background (e.g., healthy tissue) and, most importantly, from one another. Intuitively, compact clusters of higher-probability voxels in $\mathbf{S}_\theta$ are clear indicators of the model's degree of confidence about the potential existence of a lesion in the region. The first step towards lesion counting thus consists in spatially partitioning the voxels $\{v_1, v_2, ..., v_N\}$ constituting the segmentation map $\mathbf{S}_\theta$ into disjoint candidate regions $\mathcal{R}_k$ in such a way that each region contains at most one of such clusters (**Assumption 1**).

While many alternatives exist to perform this spatial clustering (Han and Uyyanonvara, 2016; Ibrahim and El-kenawy, 2020), we opt for an approach prevalent in medical instance segmentation (Wang et al., 2016; Han, 2017; Nakao et al., 2018; Cui et al., 2019), namely identifying connected components on the segmentation map $\mathbf{S}_\theta$ after binarization. In contrast to standard approaches, we suggest performing the binarization with a low threshold $\tau$ to avoid discarding any potential lesions. Of course, any prior knowledge about what constitutes a valid lesion—e.g., minimum size (Baumgartner et al., 2021; Commowick et al., 2021)—can be utilized at this stage to refine the selection of lesion candidates.

Overall, this first step of our approach maps the segmentation map $\mathbf{S}_\theta$ to a set of disjoint candidate regions $\{\mathcal{R}_1, \mathcal{R}_1, ..., \mathcal{R}_K\}$.

#### 3.1.2. Lesion Candidates → Lesion Existence Probabilities

The sets $\mathcal{R}_k$ represent regions of the image that *might* contain a lesion—according to the segmentation model $f_\theta$. Since the existence of a lesion within these clusters is not certain, we assign to each region $\mathcal{R}_k$ an underlying random variable $R_k$ indicating whether it actually

contains a lesion. In terms of modeling, the lesion-level existence probabilities $p(R_i\!=\!1)$ are intrinsically linked to the lower-level voxel probability estimates $p_\theta(V_i\!=\!1)$ outputted by the segmentation model. Indeed, a cluster of voxels that all have low probabilities is inherently less likely to contain a lesion than a region with voxel probabilities closer to 1. In fact, Bayes Theorem captures the relationship between these random variables:

$$p(V_i\!=\!1) = \frac{p(V_i\!=\!1|R_k\!=\!1)}{p(R_k\!=\!1|V_i\!=\!1)}p(R_k\!=\!1). \tag{1}$$

This equation can be simplified further in many scenarios. For instance, it is trivial that if a voxel $v_i$ is part of a lesion, then the region $\mathcal{R}_k$ to which it is assigned contains a lesion, i.e., $p(R_k = 1|V_i = 1) = 1$, if $v_i \in \mathcal{R}_k$. As a result, based on equation 1, each voxel probability $p(V_i\!=\!1)$ within a given region $\mathcal{R}_k$ is simply a linear function of the conditional $p(V_i\!=\!1|R_k\!=\!1)$ which measures the likelihood of a voxel $v_i\!\in\!\mathcal{R}_k$ to be found within the lesion's boundaries given that a lesion truly exists in $\mathcal{R}_k$. In many applications, there will be at least one voxel per cluster (generally, the centremost voxel) with a very low probability of residing outside the lesion boundaries if the cluster actually represents a lesion. Given this, the following approximation can be made (**Assumption 2**): $\max_{i:v_i\in\mathcal{R}_k} p(V_i\!=\!1|R_k\!=\!1) \approx 1$.

When combined, these results yield a segmentation-consistent estimate of the lesion existence probability of each region candidate $\mathcal{R}_k$:

$$p_\theta(R_k\!=\!1) = \max_{i:v_i\in\mathcal{R}_k} p_\theta(V_i\!=\!1). \tag{2}$$

All in all, this demonstrates that, under certain assumptions (full discussion in Appendix A), a lesion's probability of existing is best estimated by the maximum probability attributed to a voxel within the cluster. This matches the most prevalent heuristic used to aggregate individual voxel probabilities into component-level values (Han, 2017; de Moor et al., 2018; Nakao et al., 2018; Chen et al., 2019; Jaeger et al., 2020; Baumgartner et al., 2021).

### 3.1.3. Lesion Existence Probabilities → Lesion Count

So far, the segmentation map has been partitioned into disjoint lesion candidate regions $\mathcal{R}_k$ and each region has been assigned a segmentation-consistent lesion existence probability $p_\theta(R_k\!=\!1)$. By assuming that the voxels of a given region are independent of the existence of lesions within other regions (i.e., $V_i \perp\!\!\!\perp R_k, \forall i \in \{i \mid v_i \notin \mathcal{R}_k\}$), the lesion existence probabilities can be aggregated into a coherent count distribution by modeling the total number of lesions as a sum of independent Bernoulli trials (**Assumption 3**):

$$C_\theta = \sum_k \mathcal{B}(p_\theta(R_k\!=\!1)) = \sum_k \mathcal{B}(\max_{i:v_i\in\mathcal{R}_k} p_\theta(V_i\!=\!1)). \tag{3}$$

In this setup, the lesion count $C_\theta$ follows a Poisson-binomial distribution (Wang, 1993) and thus the value of its bins can be computed efficiently in quadratic time (i.e., $\mathcal{O}(K^2)$) using a simple recursion formula (Howard, 1972; Gail et al., 1981; Schroeter et al., 2019).

In fact, this instance aggregation approach and its independence assumption are reminiscent of the classical noisy-OR model (Pearl, 1988) that finds wide application in probabilistic disease modeling (Oniśko et al., 2001; Anand and Downs, 2008; Liao et al., 2019).

In summary, the proposed lesion counting approach is intuitive: it consists in identifying potential lesions (Section 3.1.1), assigning them a probability of existence (Section 3.1.2), and finally aggregating these values into a coherent count distribution (Section 3.1.3).

### 3.2. Backpropagating Count Information

The proposed Poisson-binomial lesion counting function is comprised solely of continuously differentiable operations. As a result, count information can easily be backpropagated through the function, and thus lesion count annotations $c$ can be leveraged to train the model parameters $\theta$ in addition to standard segmentation labels.

To that end, a *loss function* comparing the count label $c$ with the estimated count distribution $C_\theta$ has to first be defined. The standard Kullback-Leibler divergence (Kullback and Leibler, 1951) between the estimated count distribution $C_\theta$ and the Dirac distribution of the count labels $\mathbb{1}_c$ is a straightforward and effective choice in this setting, i.e.,

$$\mathcal{L}(\theta) = D_{KL}(\sum_k \mathcal{B}(\max_{i:v_i \in \mathcal{R}_k} p_\theta(V_i=1)) \,\|\, \mathbb{1}_c) = -\log(\,p(C_\theta = c \mid \mathbf{X})). \qquad (4)$$

This expression is equivalent to the standard cross-entropy loss function where the bins of the estimated count distribution are viewed as independent classes.

**Gradient Sparsity**  Although theoretically sound, the aggregation of voxel probabilities into a few lesion existence probabilities using a lesion-wide max-operation has for effect to produce sparse gradients. Indeed, only a tiny fraction of the voxels constituting the segmentation map (i.e., the maximum of each lesion candidate) might ultimately have non-zero gradient when backpropagating count information. While gradient sparsity can sometimes harm the learning process (Henderson and Ferrari, 2016), experiments in Section 4.2 will show quite the contrary thanks to the high correlation between neighboring voxels.

**Segmentation Pre-training**  Lesion counts are a weak source of information: they do not hold any information about the appearance, shape, or location of lesions. In terms of training, the quality of the signal provided by count labels is thus intrinsically bounded by the knowledge already assimilated by the segmentation model. For instance, the count-based gradients of a randomly initialized segmentation model can be expected to be extremely noisy, if not misleading. A stable learning procedure will therefore necessitate pre-training of the model using segmentation annotations.

## 4. Experiments and Results

Gadolinium-enhancing (Gad) lesions have been shown to be an important biomarker of disease activity in Multiple Sclerosis (McFarland et al., 1992). In this section, we thus assess the effectiveness of our approach on the clinically-relevant task of Gad lesion counting.

**Dataset**  We use a large multi-centre, multi-scanner proprietary dataset comprised of MRI scans from 1067 patients undergoing a clinical trial to treat Relapsing Remitting MS as described in (Karimaghaloo et al., 2016). The input data and training labels consist of five 3D brain MRI sequences (e.g., T1-weighted and T1-weighted with gadolinium contrast agent) and expert Gad lesion masks respectively. Patients were split into non-overlapping train (60%), validation (20%) and test (20%) sets. (See Appendix B.2 for details.)

**Baseline Segmentation Model**  As backbone for lesion segmentation, we use a 5-layer UNet (Ronneberger et al., 2015; Isensee et al., 2018) with instance normalization, leakyReLU activation, and dropout trained using Adam and cross entropy loss. (See code for details[1].)

---

1. https://github.com/SchroeterJulien/MIDL-2022-Segmentation-Consistent-Lesion-Counting

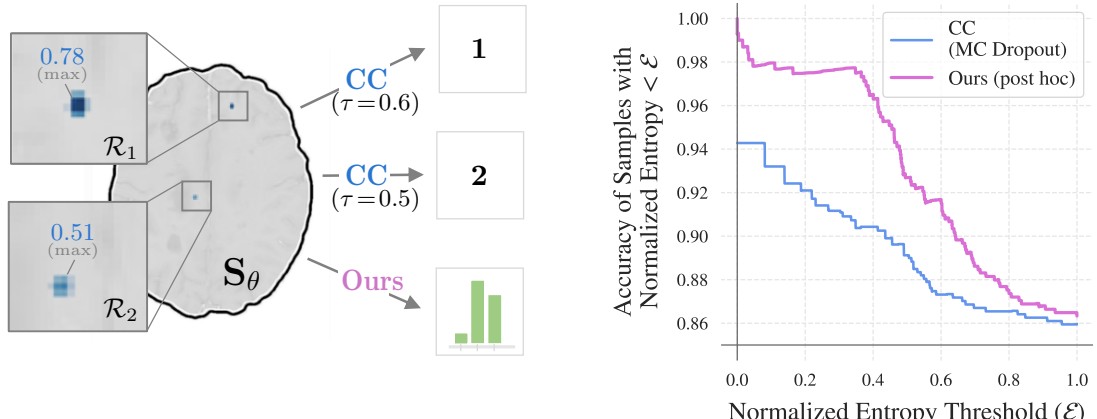

Figure 2: Post-hoc results. (**left**) Threshold $\tau$ sensitivity. Count predictions for test sample #286 with two potential lesions; CC infers a threshold-dependent scalar count, while our method infers a count distribution. (**right**) Quality of the uncertainty estimate. Overall, the more certain (lower normalized entropy) our method is, the more accurate it is.

### 4.1. Post hoc Lesion Counting

Our model can be applied directly to the segmentation output of a trained segmentation model as a post hoc counting function during inference. In this section, we compare the effectiveness of our approach against the standard connected component-based method (see Section 2). To do so, the baseline segmentation model—described above—is first trained during 115 epochs. The segmentation outputs of the test samples are then mapped to count predictions either by counting the number of connected components on a binarized version of the segmentation outputs (CC) or by using our counting function (OURS).

**Counting Performance**    While both methods exhibit comparable performance (i.e., count accuracy of around 90%) for higher binarization thresholds $\tau$, experiments reveal that our model is much more *robust* to variations in this hyperparameter (full quantitative results in Section C.1). Indeed, the count accuracy of our approach appears almost constant, while the performance of CC degrades significantly for lower thresholds (e.g., 69% accuracy at $\tau = 0.1$). Above all, the sensitivity of the connected component benchmark underlines the high fragility of its count predictions as illustrated in Figure 2 (left) where a small change in threshold leads to a drastically different count prediction. In contrast, the count distribution inferred by our model for the same example captures well the ambiguity present in the segmentation map (here, most likely one lesion, potentially two). Overall, the probabilistic nature of our approach allows to extract more information from the segmentation map, while making the predictions implicitly more robust to the binarization threshold.

**Uncertainty Conveyance**    Uncertainty quantification is key in the medical domain. We thus measure the entropy of all count distributions inferred by our model. Since scalar predictions do not intrinsically convey any uncertainty information, we use Monte Carlo dropout (Gal and Ghahramani, 2016) to estimate a comparable metric for the connected-component benchmark. This process does however not yield fully satisfactory results since the dropout-based sampling does often not produce large enough changes in the segmen-

|  | Counting | | Segmentation | | | Robustness | |
|---|---|---|---|---|---|---|---|
| METHOD | ACC | F1 | F1 | ECE | MCE | SR | AD |
| BOTTLENECK | 87.6 | 67.3 | **69.3** | 8.05 | 0.363 | 86.1 | 74.76 |
| MULTI-HEAD | **88.7** | **70.8** | 68.4 | 3.64 | 0.213 | 24.7 | 21.80 |
| DOWNSTREAM | 86.7 | 64.2 | 62.0 | 9.51 | 0.160 | 14.4 | 12.43 |
| **Ours** (Multi-task) | 88.3 | **70.8** | 66.2 | **0.72** | **0.077** | **8.1** | **7.07** |

Table 1: Performance comparison of our model and benchmarks in the multi-task setup. Reported metrics: **Counting:** accuracy (ACC) and F1-score (F1); **Segmentation:** voxel-wise F1-score (F1), expected calibration error (ECE) $[\times e^{-5}]$ and maximum calibration error (MCE) as defined in (Guo et al., 2017); **Robustness** ($\lambda = 0.5$): proportion of attacks that successfully alter the count prediction (SR) and drop in count accuracy caused by the attacks (AD).

tation map to alter the count estimate (i.e., most samples ($\sim$80%) always yield the same prediction despite the stochastic sampling). For comparison, we look at the evolution of both models' accuracy as the samples with the highest entropy are discarded (Figure 2 (right)). Most importantly, the monotone decreasing trend which can be observed demonstrates that the count distribution inferred by our model implicitly contains meaningful information about the uncertainty associated with a prediction. In addition, at very low entropy thresholds (i.e., when keeping only the most certain samples) our method reaches near 100% accuracy indicating that the model does not suffer from overconfidence, unlike CC. (These results are corroborated by the probability calibration analysis presented in Appendix C.2.)

### 4.2. Count-based Learning

Our function can also be used as a *layer* for backpropagating count information into segmentation models in a multi-task learning setting. To assess its effectiveness against standard benchmarks, we adapt three common multi-task classification and segmentation models to the task of lesion counting and segmentation. For fairness, all models are built on top of the baseline segmentation model presented above, and hence only differ architecturally in the representations used as input to perform the count classification: (BOTTLENECK) bottleneck representations (Wang et al., 2018); (MULTI-HEAD) representations taken after the second to last layer of a UNet (Mlynarski et al., 2019); (DOWNSTREAM) segmentation maps as representations (Al-Masni et al., 2020). The same training protocol is introduced for all models: pre-training for 20 epochs on segmentation labels only before training in a multi-task setup for another 100 epochs using count-based gradients with gradually increasing weight and using the validation segmentation F1-score as stopping criterion. (See Appendix B.3 for architecture and training details.)

**Multi-task Performance** Our model and the multi-head benchmark display the best overall counting metrics (see Table 1). Moreover, all multi-task approaches—except the bottleneck one—improve the calibration (Guo et al., 2017) of the segmentation predictions in comparison to the baseline segmentation model (MCE: 0.234) revealing that lesion counting acts as a relevant regularizer for segmentation learning when most parameters are shared. Finally, the 80% improvement in ECE of our model over the next best method underlines the merit of coupling the count estimate to the segmentation map for increased consistency.

**Low Data Regimes** Data scarcity is ubiquitous in medical imaging. To evaluate all models in a low data setting, we replicate the multi-task experiments using only 10% of the training samples. In this setup, all benchmarks—unlike our method—exhibit crippling convergence issues. Remarkably, applying our counting function post hoc on the output of the baseline segmentation model outperforms all approaches on all metrics (see Table C.2, in Appendix C.4), underlining the relevance of less parametrized models in low data regimes.

**Robustness** The addition of a small undetectable noise to the input *shouldn't* affect the predictions of a model intended for medical application. To assess the robustness of all models, we thus perform the following adversarial attack (Szegedy et al., 2013; Goodfellow et al., 2014) on the count predictions: 1) we freeze the models, 2) select an input sample and add a small random noise to it, 3) optimize the noise through gradient descent with the objective to fool the network (i.e., maximize entropy). We also include an $l_2$-regularization (with weight $\lambda$) in an attempt to keep the input noise as small as possible.

Overall, our approach exhibits the strongest robustness to the adversarial attack with the downstream model trailing slightly behind (see Table 1). Since the added noise is negligible, the segmentation prediction is often unaffected, and as a result the attack can be interpreted as an attack on the *consistency* between the count and segmentation outputs. This confirms the intuition that more highly parameterized models can more easily be misled to output contradictory predictions. In contrast, by coupling the count prediction to the segmentation map, our approach ensures greater consistency between the count estimate and the visual feedback provided by the segmentation, and thus is more trustworthy.

## 5. Discussion and Conclusion

In this work, we proposed a novel probabilistic model for lesion counting which can either be applied post hoc to augment segmentation models with counting capabilities or be integrated directly into the learning process in a multi-task setting. As shown through experiments on GAD lesion counting, our method outshines the commonly-used connected-component benchmark by providing well calibrated probability distributions that implicitly capture the uncertainty of the predictions. In the multi-task setup, our approach not only achieved strong counting performance but, above all, displayed the best noise robustness. Since trustworthiness and reliability are key in medical applications, our method thus offers a strong alternative to existing models which can often be fooled by minor variations of the input.

Performance aside, our model has the unique characteristic of inferring counts that can be easily traced back to precise regions of the segmentation map. The novelty of directly coupling the count estimate to the segmentation map thus grants valuable visual feedback and interpretability for clinical review—a level of transparency that is hardly achievable for more parametrized models even with the help of additional heuristics such as Grad-CAM (Selvaraju et al., 2017). However, the underlying hard consistency between count and segmentation which confers our model great robustness and interpretability is also its Achilles heel. The counting method is only as strong as the segmentation model since the count predictions are designed to faithfully reflect the content of the segmentation map and not to overturn its content (i.e., segmentation-consistent counting).

In summary, our counting approach offers a good balance between counting capabilities, segmentation consistency, performance in low data regimes, robustness, and interpretability.

## Acknowledgments

This investigation was supported by an award from the International Progressive MS Alliance (PA-1412-02420), the Canada Institute for Advanced Research (CIFAR) Artificial Intelligence Chairs program (Arbel) and by the Canadian Natural Science and Engineering Research Council (CGSM-NSERC-2021-Myers-Colet). Supplementary computational resources were provided by Calcul Québec, WestGrid, and Compute Canada. The authors would also like to thank Brennan Nichyporuk, Justin Szeto, Kirill Vasilevski, and Eric Zimmermann.

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

## Appendix A. Discussion of Model Assumptions

It is important to understand the assumptions underlying our model to better grasp its *limitations* and scope of application.

**(Assumption 1)** Our approach relies on the assumption that the lesion candidates can be identified from one another. Indeed, in our model, each lesion candidate can only contribute a maximum of 1 to the final count. For instance, the model will implicitly consider two lesions that fall within the same candidate region as one. While this assumption is automatically met in applications where lesions are *sparsely* scattered in space (e.g., lung nodules (Armato III et al., 2011), skin cancer (Yu et al., 2016; Khan et al., 2021, 2020), Gad lesions (Karimaghaloo et al., 2016)), it can be violated in applications with almost intertwined lesions (e.g., T2-weighted lesions (Salem et al., 2018)). In such case, the clustering can however be adapted to better capture the users definition of what constitutes one distinctive lesion.

**(Assumption 2)** The derivation of the lesion existence probabilities (Equation 2) introduced the assumption that, for each lesion candidate, there exists individual voxels that have no uncertainty about belonging to the lesion if it actually exists. Overall, this simplification captures well enough the situation in most standard lesion segmentation applications (e.g., brain tumors (Menze et al., 2014), lung nodules (Armato III et al., 2011), skin cancer (Yu et al., 2016; Khan et al., 2021, 2020), Gad lesions (Karimaghaloo et al., 2016), new T2-weighted lesions (Commowick et al., 2021)). This assumption can be more significantly violated in rare applications with extremely thin or tiny lesions. Indeed, in such settings, all voxels of a lesion—even the centermost ones—might be at the edge of the lesion, and thus all voxels might be suject to boundary uncertainty (i.e., the question "is the voxel outside of the lesion boundary if the lesion actually exists?" cannot be answered definitively).

**(Assumption 3)** Our approach also assumes that the voxel probabilities only depend on their assigned regions (i.e., $V_i \perp\!\!\!\perp R_k, \forall i \in \{i \mid v_i \notin \mathcal{R}_k\}$). This condition is rather weak in applications that fulfill Assumption 1. Indeed, if lesions are sparsely scattered in space then there is no ambiguity about the assignment of high probability voxels to candidate regions (i.e., no hesitation about assigning a voxel to two different regions). In such setups, the assumption weakens to:

$$R_i \perp\!\!\!\perp R_k, \forall i \neq k. \tag{5}$$

In other words, this assumption states that the existence of lesions are independent from one another, e.g., if the existence of a lesion in $\mathcal{R}_i$ came to be known, then it would have no impact on our assessment of $\mathcal{R}_k$. While it is difficult to determine the extent of the validity of this assumption in practice, this condition is at the center of the classical noisy-OR model, e.g., lung nodules (Liao et al., 2019).

As with any model, shifts from these underlying assumptions have to be considered with care, but they do not necessarily prevent the use of this approach.

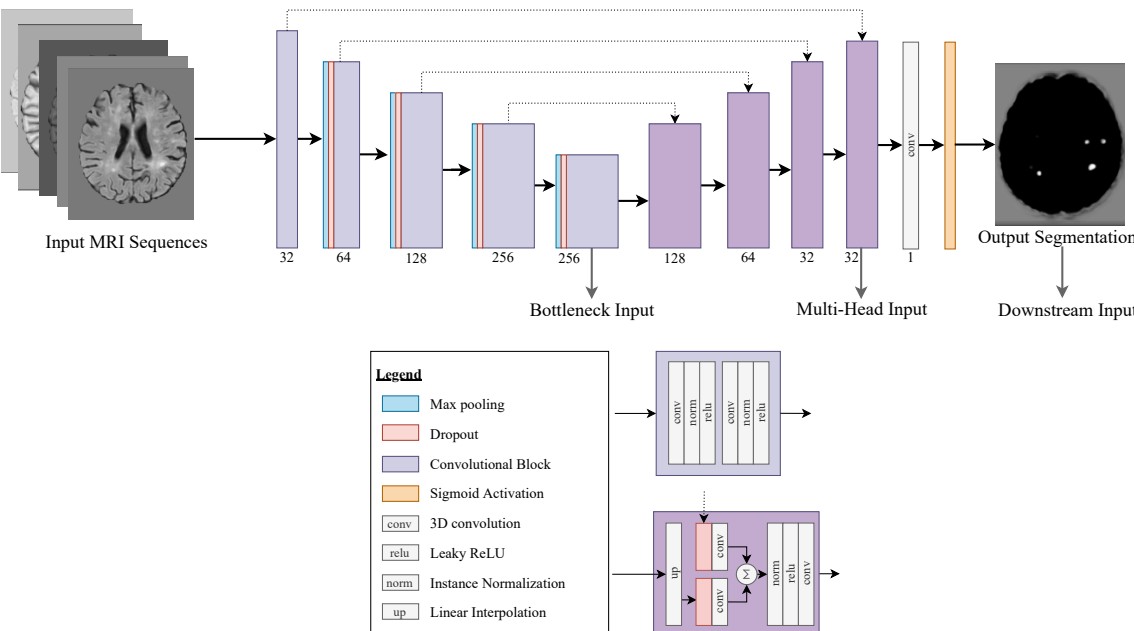

Figure B.1: Baseline segmentation UNet architecture. Dotted lines represent skip connections. Numbers underneath convolutional blocks indicate the number of filters at each level. The downward facing arrows show the representations used as input for the counting benchmarks (i.e., bottleneck, multi-head, downstream).

## Appendix B. Experiment Specifications

### B.1. Model Architecture and Training

The architecture of the segmentation model is depicted in see Figure B.1. Overall, it follows the standard UNet paradigm (Ronneberger et al., 2015; Isensee et al., 2018). The model was trained with a binary cross entropy loss with a weight of 3.0 given to the positive (lesion) class to account for class imbalance without destabilizing training. We consider count as a classification problem, and thus use count bins for training (i.e., 0, 1, 2, 3 and 4+). The Adam optimizer was used with a learning rate of $1e^{-4}$, a weight decay factor of $1e^{-5}$, $\epsilon = 1e^{-8}$ and $(\beta_1, \beta_2) = (0.9, 0.999)$. An exponential scheduler was used with a decay of 0.995. (Exceptionally for the multi-head benchmark, a learning rate of $1e^{-5}$ for the count head parameters was needed to avoid convergence issues.) All benchmark counting models were trained using the following scheme: we first pre-train a segmentation UNet for 20 epochs as required by Section 3.2 after which we weigh the counting loss using $\alpha e^{\beta(\text{epoch}-20)}$ with $\alpha = 0.001$ and $\beta = 0.06$ which were chosen through trial and error. Models are selected based on validation segmentation F1 score.

### B.2. Dataset

Our dataset consists of MRI scans from 1067 patients undergoing a clinical trial to treat Relapsing Remitting Multiple Sclerosis (RRMS). Each patient was monitored for up to three years with annual scans. We consider one scan to be a single data sample for our

| Threshold | | $\tau = 0.1$ | 0.2 | 0.3 | 0.4 | 0.5 | 0.6 | 0.7 | 0.8 | 0.9 |
|---|---|---|---|---|---|---|---|---|---|---|
| **Accuracy** | CC | 69.2 | 75.7 | 80.7 | 83.6 | 85.8 | 87.6 | 90.3 | 90.1 | 86.2 |
| | Ours | 86.3 | 86.3 | 86.5 | 86.5 | 86.0 | 87.8 | 90.3 | 90.1 | 86.2 |
| **F1-score** | CC | 54.2 | 60.5 | 64.5 | 68.4 | 71.4 | 73.7 | 78.1 | 75.3 | 65.4 |
| | Ours | 74.1 | 74.2 | 74.5 | 74.5 | 72.7 | 74.2 | 78.1 | 75.3 | 65.4 |
| **Precision** | CC | 50.9 | 57.0 | 60.9 | 65.2 | 68.5 | 71.6 | 77.0 | 76.1 | 70.1 |
| | Ours | 71.7 | 71.9 | 72.3 | 72.1 | 69.9 | 72.2 | 77.0 | 76.1 | 70.1 |
| **Recall** | CC | 61.9 | 66.9 | 69.9 | 73.0 | 75.4 | 76.3 | 79.5 | 74.6 | 62.5 |
| | Ours | 77.7 | 77.7 | 78.2 | 78.2 | 76.3 | 76.5 | 79.5 | 74.6 | 62.5 |

Table C.1: Performance sensitivity to binarization threshold $\tau$. Test accuracy, class-average F1-score, precision and recall are reported. Note that across metrics, our method is stable.

network. To ensure no patient appears in more than one dataset, we split by patient before separating samples into annual scans. Thus, a patient may appear up to three times in one set, depending on the number of annuals scans they underwent. The Gad lesion masks were independently produced by two raters who then agree to a consensus. Per patient Gad lesion counts are derived directly from the lesion masks. We differ slightly from Karimaghaloo et al. (2016) in our preprocessing. Here, the data was padded and cropped to be $64 \times 192 \times 192$, skull-stripped, denoised and standardized. We provide our models with five MRI sequences, namely T1-weighted, T1-weighted with gadolinium contrast agent, T2-weighted, Fluid Attenuated Inverse Recovery, and Proton Density weighted.

### B.3. Benchmarks

The UNet architecture is depicted in Figure B.1 with the input to each of the multi-task networks indicated. The counting networks consist of fully convolutional networks of 3 (bottleneck) or 7 layers (multi-head, downstream). The bottleneck convolutional blocks are made up of a ReLU, a 3D convolution, an average pooling followed by a dropout layer (except in the last block before the output). Similarly, for the deeper models, each convolutional block is made up of an extra ReLU and 3D convolution layer before following the same architecture as the bottleneck blocks.

## Appendix C. Additional Results

### C.1. Threshold Sensitivity

Table C.1 shows the full results for both our method (applied post hoc on the outputs of the segmentation model) and the connected components benchmark at various thresholds $\tau$. Across all reported metrics, the connected components approach is entirely dependent on the binarization threshold in the $[0.0, 0.5]$ range where we note large changes in performance. By contrast, our method is much more stable throughout the entire range of thresholds.

This is further shown in the confusion matrices of the methods at three sample thresholds (Figure C.2). Again, our method is considerably more consistent across the different

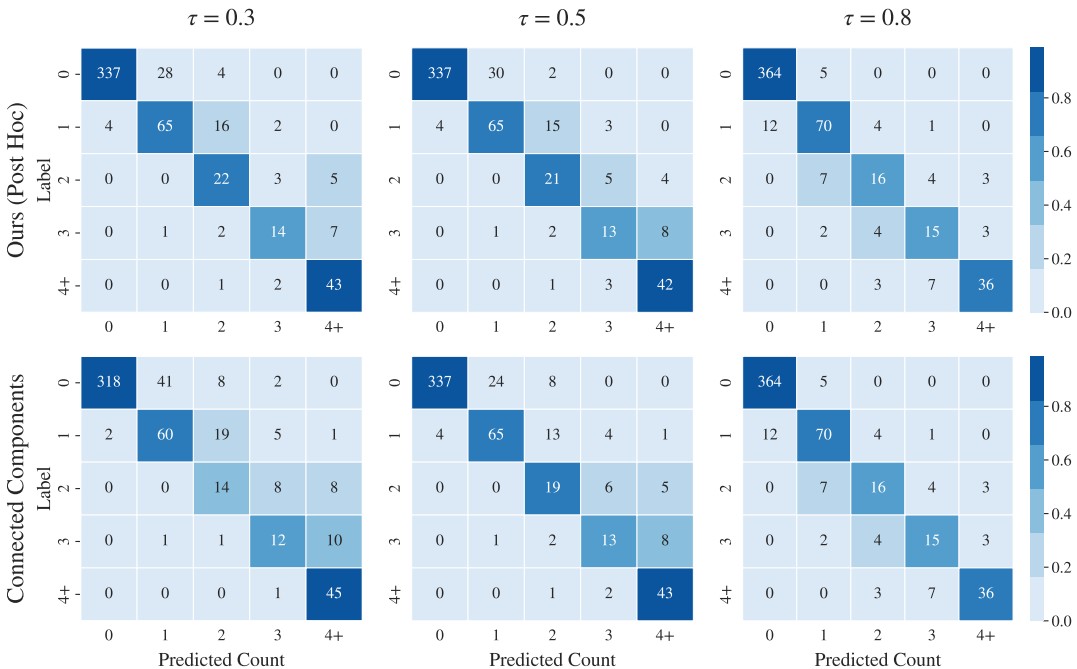

Figure C.2: Confusion matrices comparing post hoc methods at various thresholds

thresholds than connected components, whose performance decreases at a lower threshold, most notably in the confusion across classes 2, 3 and 4.

## C.2. Probability Calibration

In contrast to the connected component benchmark which outputs scalar count predictions, our method produces count distribution. The calibration curve (Guo et al., 2017) in Figure C.3 reveals that the probabilities inferred by our model are well-calibrated demonstrating, above all, the merit of the lesion-level maximum voxel aggregation derived in Section 3.1.2. It also shows that our method conveys meaningful additional information whereas the connected component has only two possible confidence values (0 or 1). For instance, in Figure 2 (left), our approach expresses the potential existence of a second lesion in the segmentation map, while the benchmark makes a hard threshold-dependent decision.

## C.3. Uncertainty Conveyance

As our method outputs a probability distribution, we can obtain an uncertainty estimate for each sample by evaluating it's entropy. To compare against connected components, we generate samples of the segmentation map using MC Dropout (Gal and Ghahramani, 2016) and evaluate the count independently on each sample to obtain a count distribution. We use a threshold of $\tau = 0.5$ for the connected components benchmark. For this experiment, we use 50 MC Dropout samples. Figure C.4 shows the result of thresholding the predictions based on entropy. When we keep only the least uncertain test samples (Figure C.4b), our

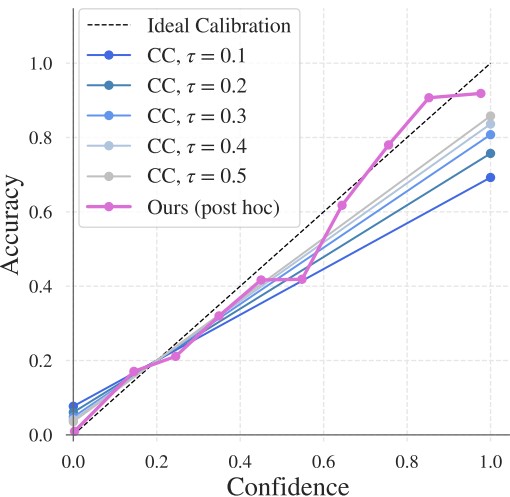

Figure C.3: Count calibration at various thresholds for CC and for our method at $\tau = 0.1$.

model achieves an accuracy of 100%, unlike the connected components approach. Furthermore, though both methods have the majority of their predictions with entropy close to 0, our method has considerably fewer samples with 0 entropy while also producing a greater variability (see figure C.5). Most notably, our method produces greater uncertainty values when it is incorrect with very few incorrect samples having low uncertainty. By contrast, the connected components approach produces a similar entropy distribution for both correct and incorrect samples. Both of these results coupled together show that our method is correct when it is certain and also maintains the ability to produce meaningful uncertainty values.

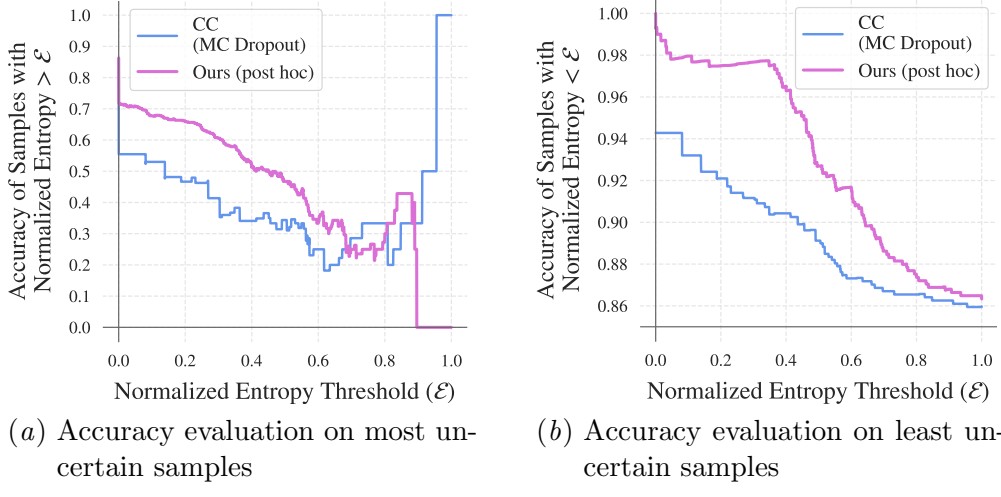

(a) Accuracy evaluation on most uncertain samples

(b) Accuracy evaluation on least uncertain samples

Figure C.4: Accuracy as a function of uncertainty.

|  | Counting | | Segmentation | | |
| --- | --- | --- | --- | --- | --- |
| METHOD | ACCURACY | F1-SCORE | F1-SCORE | ECE ($\times e^{-5}$) | MCE |
| BOTTLENECK | 76.6 | 41.5 | 66.7 | 6.13 | 0.306 |
| MULTI-HEAD | 80.8 | 47.6 | 58.0 | 5.13 | 0.276 |
| DOWNSTREAM | 82.9 | 54.4 | 59.2 | 1.64 | 0.119 |
| **Ours** (Multi-task) | 86.0 | 65.1 | 58.4 | 2.70 | 0.524 |
| **Ours** (Post hoc) | **86.5** | **70.9** | **68.2** | **0.95** | **0.116** |

Table C.2: Low data regime count and segmentation performance. Segmentation metrics are reported voxel-wise.

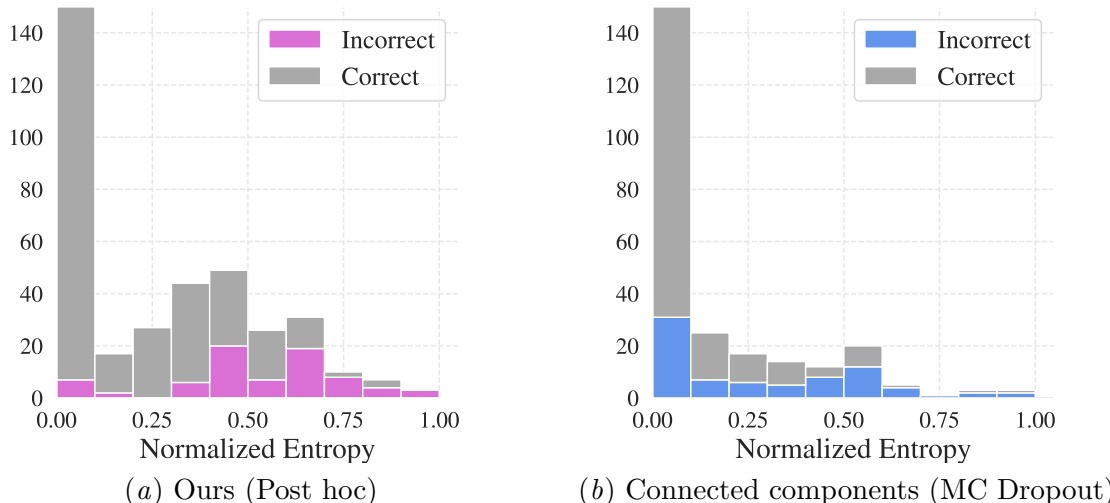

$(a)$ Ours (Post hoc) $\qquad\qquad$ $(b)$ Connected components (MC Dropout)

Figure C.5: Distribution of entropy values for correctly and incorrectly classified samples.

## C.4. Low Data Regimes

We train our models with only 10% of the training data (i.e. 167 samples) to assess their perform when data is scarce. The results are summarized in Table C.2. Our method, both post hoc and multi-task versions, achieves the highest count accuracy and F1-score.

## C.5. Adversarial Attacks

**Experiment Setup** The goal of the adversarial attack experiment is to change the count predictions outputted by the multi-task models, while making minimal changes to the input. To do so, we add a noise parameter to the inputs which we train to *maximize* the cross entropy between the label and the prediction. During the attacks, all models are frozen. For every test sample, we initialize a noise parameter with a random Gaussian noise with zero mean and a standard deviation of $1e^{-5}$. We use the Adam optimizer (Kingma and

| $l_2$-reg. | | Accuracy After | Accuracy Drop | Success Rate |
|---|---|---|---|---|
| $\lambda = 0.1$ | Bottleneck | 2.3 | 84.51 | 97.4 |
| | Multi-Head | 64.8 | 23.33 | 26.5 |
| | Downstream | 65.4 | 20.65 | 24.0 |
| | **Ours** | **75.1** | **12.62** | **14.4** |
| $\lambda = 0.5$ | Bottleneck | 12.0 | 74.76 | 86.1 |
| | Multi-Head | 66.3 | 21.80 | 24.7 |
| | Downstream | 73.6 | 12.43 | 14.4 |
| | **Ours** | **80.7** | **7.07** | **8.1** |
| $\lambda = 0.9$ | Bottleneck | 17.8 | 69.02 | 79.5 |
| | Multi-Head | 67.5 | 20.65 | 23.4 |
| | Downstream | 77.1 | 8.99 | 10.4 |
| | **Ours** | **83.9** | **3.82** | **4.4** |

Table C.3: Full adversarial attack results on all multi-task models. Success rate is calculated as percentage of attacks that successfully altered the prediction

Ba, 2014) with a learning rate of $1e^{-4}$. We limit our attacks to samples with a count label of 4 or less. This is because our last classification bin includes all counts from 4 to 65 (the maximum count in the dataset). For large counts, it would not be practical to attempt to change the prediction as it would need to decrease to a count of 3. Nevertheless, we include results on 523 out of 556 test samples. Finally, we train until the count prediction no longer matches the label (success) or until 500 epochs have passed (failure).

**Regularization** In order to limit the changes made to the input and to the resultant segmentation map (i.e. we do not want to artificially add a lesion and penalize the model for increasing the count), we add an $\ell_2$-regularization on the change in the segmentation map. This regularization is weighted by parameter $\lambda$ (see Table C.3). Furthermore, without regularization, the noise parameter is unbounded and could potentially create meaningless inputs and by extension incorrect count predictions on all models. It is therefore important to keep the additive noise as small as possible to ensure the ground truth label remains true. As can be seen in Figure C.6, both the noise itself and the change in the segmentation prediction remains small.

**Results** We run our adversarial attacks with several levels of regularization. As can be seen in Table C.3, our method is the least susceptible to this type of adversarial attack regardless of the amount of regularization.

Figure C.6 shows exemplary samples that have been altered by an adversarial attack to change the predicted count. We display the subtraction of the post-contrast and pre-contrast T1-weighted MRI both before and after the adversarial attack. Note the subtlety of the changes which have been applied. The samples for each model were chosen using the same criterion: (1) calculate the maximum change in the segmentation map and (2) select the sample with the smallest maximum. On a per sample basis, the maximum discrepancy in the segmentation map indicates what the adversarial attack targeted. However, by selecting the image with the minimum discrepancy, we can show how easily each model was fooled.

Since our method allows us to map from count to segmentation, we know *exactly* which voxels are responsible for the change in count prediction. As shown in Figure C.6 (bottom right), a slight increase in the depicted voxel probability resulted in an increased count prediction. Ours is the only method to offer this level of transparency.

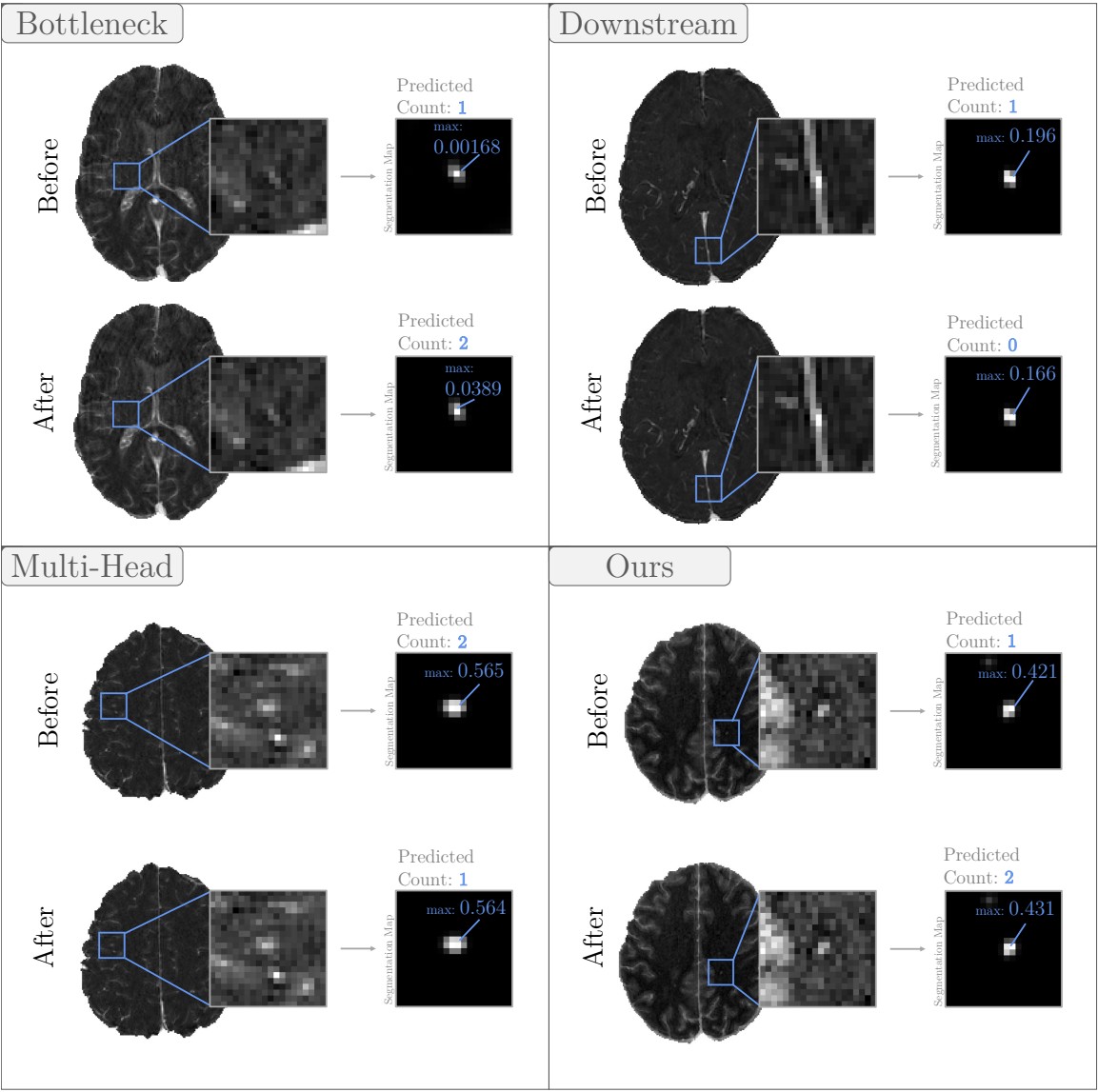

Figure C.6: Examples of adversarial attacks performed on each of the models. Depicted is both the input and segmentation map before and after the addition of adversarial noise.

