# OpenReview forum: "Segmentation-Consistent Probabilistic Lesion Counting"
_MIDL.io/2022/Conference — MIDL 2022_

### Official Review · Reviewer_FbgQ · 2022-01-19

**Confidence:** 4
**Preliminary Rating:** 3
**Recommendation:** Poster

**Summary:**

The authors propose a differentiable mapping function from lesion segmentation mask to lesion count distribution to enable end-to-end lesion counting training. They use Bayes theorem to approximate each individual lesion probabilities and aggregate them into a joint count distribution with a Poisson-binomial distribution by assuming that the existence of each lesion is independent. Experiments have been performed on a large scale Gad lesion dataset and the proposed method outperforms traditional method in post-hoc setup. In the count-based learning setup, the proposed method shows better calibration performance compared to the baseline methods.

**Strengths:**

1. The problem is well-motivated that lesion counts information couldn’t be used for supervising in deep learning due to the discontinuity of the integers.
2. The proposed method is based on many domain-specific knowledge and the mathematical derivations are valid.
3. A large-scale dataset has been used for performance evaluation.


**Weaknesses:**

1. Although in the post-hot counting setup the proposed method outperforms the traditional one, it’s not a deep learning approach where the mapping function is non-parametric which does not fit the scope of MIDL.
2. In the count-based learning setup, the performance of the proposed method is similar to the multi-head one.


**Deanonymize Review:**

no

**Final Rating After The Rebuttal:**

5: Strong Accept

**Justification Of The Final Rating:**

In general, I'm happy with the response which addressed my questions. Although there were ordering problems in text and experimental results, this is a solid paper with both mathematical insights and abundant experiments. I would like to recommend for acceptance.

**Paper Type:**

methodological development

**Questions To Address In The Rebuttal:**

1. By leveraging the gradient from lesion counting function for model updates, the count-based learning setup falls in the category of DL. However, the count performance of the proposed method is similar to the multi-head one in Table C.2.
2. Although the proposed method helps in segmentation calibration, the focus of the paper is on lesion counting.
3. A question about the three benchmarks that have been used in count-based learning. The authors used words “similar to”, “inspired by”, and “like in” from line 212 to line 215 to describe the benchmarks. A natural doubt is that are the baseline methods you compared with the original ones?


**Special Issue:**

no

---

### Official Review · Reviewer_HtKU · 2022-01-24

**Confidence:** 3
**Preliminary Rating:** 2
**Recommendation:** Poster

**Summary:**

The paper presents a continuous and differentiable function that maps lesion segmentation prediction to lesion count probability. The main motivation is that normal lesion counting (connected-component-based) is not differentiable and therefore cannot be used during training of a neural network. The authors perform quite a lot of experiments on a Gadolinium-enhancing (Gad) lesions MRI dataset.

**Strengths:**

- The method description is easy to follow and written nicely.
- The code is available.
- The paper gives a very detailed introduction to previous works.
- The authors performed a large number of experiments to give insights into the method

**Weaknesses:**

Due to the large number of experiments, no results are presented detailed enough in the paper itself. To understand the results and the paper, it is necessary to read most of the appendix (that is as long as the paper itself).
-	Additionally, the focus of the paper gets lost due to all these experiments.
-	The discussion including limitations of the presented method is completely missing.
-	No examples are shown in a figure demonstrating the results and cases in which one or the other method performs better.
-	The captions of the figures are not sufficient to understand the figures (especially not without the corresponding description in the text)

-	Even after reading the paper, I do not understand why the benefit of the proposed method (maybe because of the paper structure?). For a threshold of \tau = 0.5, it doesn’t make a difference which counting method I am using, so why should I use the more complex one? The segmentation networks are trained with \tau =0.5 or expect that and if it works with that, why do I want to be more robust against that parameter? Also, for the multi-task learning, I don’t see a benefit of the proposed method. Maybe there is one, but then please highlight it more and focus on it!

Open Questions:
-	What happens if only one voxel has a high probability in a cluster? That would mean that R_i = 1, but a lesion consists of more than one voxel.
-	Does assumption 3 really holds? The existence of lesions is not independent..


**Deanonymize Review:**

no

**Detailed Comments:**

Please rename the "Experiments" section to "Experiments and Results" section.

**Final Rating After The Rebuttal:**

4: Weak Accept

**Justification Of The Final Rating:**

Thank you for the detailed answer.

I appreciate the updates the authors made. They streghthen the presented paper and make it easier to follow.

Now I am happy to recommend accepting the paper. I am looking forward to the presentation at MIDL.


**Paper Type:**

methodological development

**Questions To Address In The Rebuttal:**

My main concern about this paper is that due to the large number of experiments, the focus is missing and the paper gets unstructured. Please focus on the main finding and leave out unnecessary experiments like the adversarial attack experiment (I don’t understand for what that is important for medical images and if it is that should be a different paper or you can add such an experiment completely in the appendix) or the low data regimes (interesting experiment but the focus of the paper should be somewhere else!)
Moreover, the paper needs a proper discussion of the results. What are the weaknesses on which cases does the one or the other method work better? How are the results related to the related work section? (This discussion can be added to the “Conclusion” section that should be renamed than to “Discussion and Conclusion

**Special Issue:**

no

---

### Meta-Review · Area_Chair_QEgH · 2022-02-20

**Recommendation:** Accept (Poster)
**Confidence:** 4

**Metareview:**

The authors have addressed the questions raised by the reviewers and have clearly improved the manuscript.

---

### Decision · Program_Chairs · 2022-02-28

Accept